# Discovering Motor Programs by Recomposing Demonstrations

**Tanmay Shankar**
Facebook AI Research
tanmayshankar@fb.com

**Shubham Tulsiani**
Facebook AI Research
shubhtuls@fb.com

**Lerrel Pinto**
Robotics Institute, CMU
lerrelp@cs.cmu.edu

**Abhinav Gupta**
Facebook AI Research
gabhinav@fb.com

## Abstract

In this paper, we present an approach to learn recomposable motor primitives across large-scale and diverse manipulation demonstrations. Current approaches to decomposing demonstrations into primitives often assume manually defined primitives and bypass the difficulty of discovering these primitives. On the other hand, approaches in primitive discovery put restrictive assumptions on the complexity of a primitive, which limit applicability to narrow tasks. Our approach attempts to circumvent these challenges by jointly learning both the underlying motor primitives and recomposing these primitives to form the original demonstration. Through constraints on both the parsimony of primitive decomposition and the simplicity of a given primitive, we are able to learn a diverse set of motor primitives, as well as a coherent latent representation for these primitives. We demonstrate, both qualitatively and quantitatively, that our learned primitives capture semantically meaningful aspects of a demonstration. This allows us to compose these primitives in a hierarchical reinforcement learning setup to efficiently solve robotic manipulation tasks like reaching and pushing.

## 1 Introduction

We have seen impressive progress over the recent years in learning based approaches to perform a plethora of manipulation tasks (Levine et al., 2016; Andrychowicz et al., 2018; Levine et al., 2018; Pinto & Gupta, 2016; Agrawal et al., 2016). However, these systems are typically task-centric *savants* – able to only execute a single task that they were trained for. This is because these systems, whether leveraging demonstrations or environmental rewards, attempt to learn each task *tabula rasa*, where low to high level motor behaviours, are all acquired from scratch in context of the specified task. In contrast, we humans are adept at a variety of basic manipulation skills *e.g.* picking, pushing, grasping *etc.*, and can effortlessly perform these diverse tasks via a unified manipulation system.

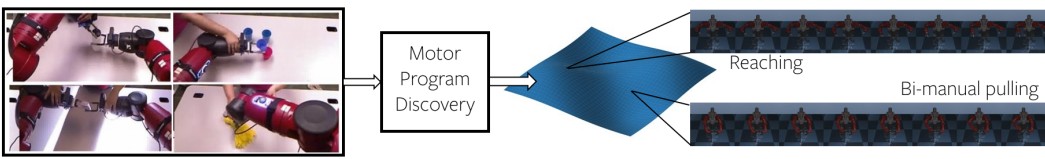

**Figure 1:** Sample motor programs that emerge by discovering the space of motor programs from a diverse set of robot demonstration data in an unsupervised manner. These motor programs facilitate understanding the commonalities across various demonstrations, and accelerate learning for downstream tasks.

How can we step-away from the paradigm of learning task-centric savants, and move towards building similar unified manipulation systems? We can begin by not treating these tasks independently, but

via instead exploiting the commonalities across them. One such commonality relates to the primitive actions executed to accomplish the tasks – while the high-level semantics of tasks may differ significantly, the low and mid-level *motor programs* across them are often shared *e.g.* to either pick or push an object, one must move the hand towards it. This concept of *motor programs* can be traced back to the work of Lashley, who noted that human motor movements consist of 'orderly sequences' that are not simply sequences of stimulus-response patterns. The term 'motor programs' is however better attributed to Keele (1968) as being representative of 'muscle commands that execute a movement sequence uninfluenced by peripheral feedback', though later works shifted the focus from muscle commands to the movement itself, while allowing for some feedback (Adams, 1971). More directly relevant to our motivation is Schmidt's notion of 'generalized' motor programs (Schmidt, 1975) that can allow abstracting a class of movement patterns instead of a singular one. In this work, we present an approach to discover the shared space of (generalized) motor programs underlying a variety of tasks, and show that elements from this space can be composed to accomplish diverse tasks. Not only does this allow understanding the commonalities and shared structure across diverse skills, the discovered space of motor programs can provide a high-level abstraction using which new skills can be acquired quickly by simply learning the set of desired motor programs to compose.

We are not the first to advocate the use of such mid-level primitives for efficient learning or generalization, and there have been several reincarnations of this idea over the decades, from 'operators' in the classical STRIPS algorithm (Fikes & Nilsson, 1971), to 'options' (Sutton et al., 1999) or 'primitives' (Schaal et al., 2005) in modern usage. These previous approaches however assume a set of manually defined/programmed primitives and therefore bypass the difficulty of discovering them. While some attempts have been made to simultaneously learn the desired skill and the underlying primitives, learning both from scratch is difficult, and are therefore restricted to narrow tasks. Towards overcoming this difficulty, we observe that instead of learning the primitives from scratch in the context of a specific task, we can instead discover them using demonstrations of a diverse set of tasks. Concretely, by leveraging demonstrations for different skills *e.g.* pouring, grasping, opening *etc.*, we discover the motor programs (or movement primitives) that occur across these.

We present an approach to discover movement primitives from a set of *unstructured* robot demonstration *i.e.* demonstrations without additional parsing or segmentation labels available. This is a challenging task as each demonstration is composed of a varying number of unknown primitives, and therefore the process of learning entails both, learning the space of primitives as well as understanding the available demonstrations in context of these. Our approach is based on the insight that an abstraction of a demonstrations into a sequence of motor programs or primitives, each of which correspond to an implied movement sequence, and must yield back the demonstration when the inferred primitives are 'recomposed'. We build on this and formulate an unsupervised approach to jointly learn the space of movement primitives, as well as a parsing of the available demonstrations into a high-level sequence of these primitives.

We demonstrate that our method allows us to learn a primitive space that captures the shared motions required across diverse skills, and that these motor programs can be adapted and composed to further perform specific tasks. Furthermore, we show that these motor programs are semantically meaningful, and can be recombined to solved robotic tasks using reinforcement learning. Specifically, solving *reaching* and *pushing* tasks with reinforcement learning over the space of primitives achieves 2 orders of magnitude faster training than reinforcement learning in the low-level control space.

## 2 RELATED WORK

Our work is broadly related to several different lines of work which either learn task policies from demonstrations, or leverage known primitives for various applications, or learn primitives in context of known segments. While we discuss these relations in more detail below, we note that previous primitive based approaches either require: a) a known and fixed primitive space, or b) annotated segments each corresponding to a primitive. In contrast, we *learn* the space of primitives *without requiring this segmentation annotation*, and would like to emphasize that ours is the first work to do so for a diverse set of demonstrations spanning multiple tasks.

**Learning from Demonstration:** The field of learning from demonstrations (LfD) (Nicolescu & Mataric, 2003) has sought to learn to perform tasks from a set of demonstrated behaviors. A number of techniques exist to do so, including cloning the demonstrated behavior (Esmaili et al., 1995),

fitting a parametric model to the demonstrations (Kober & Peters, 2009; Peters et al., 2013), or first segmenting the demonstrations and fitting a model to each of the resultant segments (Niekum et al., 2012; Krishnan et al., 2018; Murali et al., 2016; Meier et al., 2011). We point the reader to Argall et al. (2009) for a comprehensive overview of the field. Rather than directly learn to perform tasks, we use demonstrations to learn a diverse set of composable and reusable primitives, or motor-programs that may be used to perform a variety of downstream tasks.

**Learning and Sequencing Motion Primitives:** Several works (Kober & Peters, 2009; Peters et al., 2013) learn motion primitives from demonstrations using predetermined representations of skills such as Dynamic Movement Primitives (DMPs) (Schaal et al., 2005). A few other works have approached the problem from an optimization perspective (Dragan et al., 2015). Given these primitives, a question that arises is how to then sequence learned skills to perform downstream tasks. Several works have attempted to answer this question - (Neumann et al., 2014) builds a layered approach to adapt, select, and sequence DMPs. Konidaris et al. (2012); Konidaris & Barto (2009) segments demonstrations into sequences of skills, and also merge these skills into skill-trees. However, the predetermined representations of primitives adopted in these works can prove restrictive. In particular, it prevents learning arbitrarily expressive motions and adapting these motions to a generic downstream task. We seek to move away from these fixed representations of primitives, and instead learn representations of primitives along with the primitives themselves.

**Latent Variable Models:** The family of latent variable models (LVMs) provides us with fitting machinery to do so. Indeed, the success of LVMs in learning representations in deep learning has inspired several recent works (Co-Reyes et al., 2018; Kipf et al., 2019; Haarnoja et al., 2018; Lynch et al., 2019) to learn latent representations of trajectories. SeCTAR (Co-Reyes et al., 2018) builds a latent variable conditioned policy and model that are constrained to be consistent with one another, and uses the learned policies and model for hierarchical reinforcement learning. The CompILE framework (Kipf et al., 2019) seeks to learn variable length trajectory segments from demonstrations instead, and uses latent variables to represent these trajectory segments, but is evaluated in relatively low-dimensional domains. We adopt a similar perspective to these works and learn continuous latent variable representations (or abstractions) of trajectory *segments*.

**Hierarchical RL and the Options Framework:** The related field of hierarchical reinforcement learning (HRL) learns a layering of policies that each abstract away details of control of the policies of a lower level. The Options framework (Sutton et al., 1999) also learns similar temporal abstractions over sequences of atomic actions. While promising, its application has traditionally been restricted to simple domains due to the difficulty of jointly learning internal option policies along with a policy over those options. Recent works have managed to do so with only a reward function as feedback. The Option-Critic framework (Bacon et al., 2017) employs a policy-gradient formulation of options to do so, while the Option-Gradient (Smith et al., 2018) learns options in an off-policy manner. In contrast with most prior work in the options framework, (Daniel et al., 2016; Fox et al., 2017; Krishnan et al., 2017) learn options from a set of demonstrations, rather than in the RL setting. In similar spirit to these works, we too seek to learn abstractions from a given set of demonstrations, however unlike DDO (Fox et al., 2017), DDCO (Krishnan et al., 2017), and CompILE (Kipf et al., 2019), we can learn primitives beyond a discrete set of options in a relatively high dimensional domain.

**Hierarchical representations of demonstrations:** The idea of hierarchical task representations has permeated into LfD as well. In contrast to reasoning about demonstrations in a flat manner, one may also infer the hierarchical structure of tasks performed in demonstrations. A few recent works have striven to do so, by representing these tasks as programs (Xu et al., 2018; Sun et al., 2018), or as task graphs (Huang et al., 2019). Both Xu et al. (2018) and Huang et al. (2019) address generalizing to new instances of manipulation tasks in the low-shot regime by abstracting away low-level controls.

The idea of policy sketches, i.e. a sketch of the sub-tasks to be accomplished in a particular task, has become popular (Andreas et al., 2017; Shiarlis et al., 2018). Andreas et al. (2017) learn modular policies in the RL setting provided with such policy sketches. Shiarlis et al. (2018) provides a modular LfD framework based on this idea of policy sketches. While all of these works address learning policies at various levels from demonstrations, unlike our approach, they each assume access to heavy supervision over demonstrations to do so.

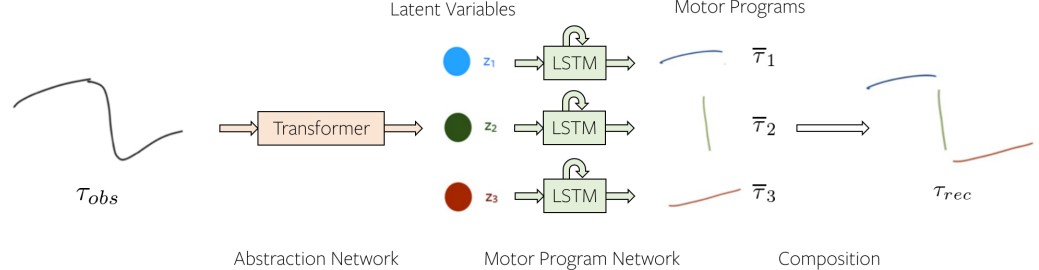

**Figure 2:** An overview of our approach. Our abstraction network takes in an observed demonstration $\tau_{obs}$ and predict a sequence of latent variables $\{z\}$. These $\{z\}$ are each decoded into their corresponding motor programs via the motor program network. We finally recompose these motor programs into the recomposed trajectory.

## 3  APPROACH

We seek to discover the space of motor programs directly from unstructured demonstrations in an unsupervised manner, and show that these can help understanding similarities across tasks as well as quickly quickly adapting to and solving new tasks. Building on ideas of Keele (1968) and Schmidt (1975), we define a motor program $M$ as a movement pattern that may be executed in and of itself, without access to sensory feedback. Concretely, a 'movement sequence' or a motor program $M$ is a sequence of robot joint configurations. Our goal is to learn the space of such movement patterns that are present across a diverse set of tasks. We do so via learning a 'motor program network' that maps elements $z \in R^n$ to corresponding movement sequences *i.e.* $\mathcal{M} : z \longrightarrow M$.

Given a set of $N$ unlabelled demonstrations $\{\tau_i\}_{i=1,2,...,N}$ that consist of sequences of robot states $\{s_1, s_2, ..., s_T\} \in S$, our aim is to learn the shared space of motor programs via the motor program network $\mathcal{M}$. However, as each demonstration $\tau_i$ is unannotated, we do not apriori know what subsequences of the demonstration correspond to a distinct motor program. Therefore, to learn motor programs from these demonstrations, we need to simultaneously learn to understand the demonstrated trajectories in terms of composition of motor programs. Thus in addition to learning the network $\mathcal{M}$, we also learn a mapping $\mathcal{A}$ from each demonstrated trajectory $\tau_i$ to the underlying sequence of motor programs $\{M_1, M_2, ..., M_K\}$ (and associated latent variables $\{z_1, z_2, ..., z_K\}$) executed during the span of the trajectory. We call this mapping $\mathcal{A} : \tau_i \longrightarrow \{M_1, M_2, ..., M_K\}$ the abstraction network, as it abstracts away the details of the trajectory into the set of motor programs (i.e., abstractions). Note that both the abstraction and motor program networks are learned using only a set of demonstrations from across diverse tasks.

### 3.1  PRIMITIVE DISCOVERY VIA RECOMPOSITION

Our central insight is that we can jointly learn the space of motor programs and the abstraction of the demonstration by enforcing that the implied 'recomposition' is faithful to the original demonstration. Concretely, an abstraction of a demonstration into a sequence of motor programs, each of which corresponds to an implied motion sequence, must yield back the original demonstration when the inferred motor programs are decoded and 'recomposed'. We operationalize this insight to jointly train the motor program and abstraction networks from demonstrations.

**Learning Overview and Objective:** Our approach is outlined in Fig 2, where given an input demonstration trajectory $\tau_{\text{obs}}$, we use the abstraction network to predict a sequence of (a variable number of) latent codes $\{z_k\}$. These are each decoded into corresponding sub-trajectories via the learned motor program network $\mathcal{M}$.

$$\{z_k\} = \mathcal{A}(\tau_{obs}); \quad \bar{\tau}_k = \mathcal{M}(z_k) \tag{1}$$

Given the decoding of the predicted motor programs, we can recompose these sub-trajectories to obtain a *recomposed* demonstration trajectory $\tau_{rec}$, and penalize the discrepancy between the observed and the recomposed demonstrations. Denoting by $\oplus$ the concatenation operator, our loss for a given demonstration $\tau_{obs}$ is therefore characterized as:

$$\tau_{rec} = \bar{\tau}_1 \oplus \bar{\tau}_2 \cdots \oplus \bar{\tau}_K; \quad L(\tau_{obs}; \mathcal{M}, \mathcal{A}) = \Delta(\tau_{obs}, \tau_{rec}) \tag{2}$$

As the trajectories $\tau_{obs}, \tau_{rec}$ are possibly of different lengths, we use a pairwise matching cost between the two trajectories, where the optimal alignment is computed via dynamic time warping (Berndt &

Clifford, 1994). This provides us with a more robust cost measure that handles different prediction lengths, is invariant to minor velocity perturbations, and enables the model to discard regions of inactivity in the demonstrations. Given two trajectories $\tau_a \equiv (s_1 \cdots s_M)$, $\tau_b \equiv (s_1 \cdots s_N)$, and a distance metric $\delta$ over the state space, the discrepancy measure between trajectories can be defined as the matching cost for the optimal matching path $P$ among all possible valid matching paths $\mathcal{P}$ (*i.e.* paths satisfying monotonicity, continuity, and boundary conditions (Berndt & Clifford, 1994)):

$$\Delta(\tau_a, \tau_b) = \min_{P \in \mathcal{P}} \sum_{(m,n) \in P} \delta(\tau_a[m], \tau_b[n]) \tag{3}$$

As the recomposed trajectory comprises of distinct primitives, each of which implies a sequence of states, we sometimes observe discontinuities *i.e.* large state changes between the boundaries of these primitives. To prevent this, we additionally incorporate a smoothness loss $L_{sm}(\tau_{rec})$ that penalizes the state change across consecutive time-steps if they are larger than a certain margin. Our overall objective, comprising of the reconstruction objective and the smoothness prior, can allow us to jointly learn the space of motor programs and the abstraction of trajectories in an unsupervised manner.

**Network Architecture and Implementation Details.** We parameterize our motor program network $\mathcal{M}$ and our abstraction network $\mathcal{A}$ as neural networks. In particular, the motor program network is a 4 layer LSTM (Graves et al., 2013) that takes a single 64 dimensional latent variable $z$ as input, and predicts a sequence of 16 dimensional states. For our abstraction network, we adopt the Transformer (Vaswani et al., 2017) architecture to take in a varying length 16 dimensional continuous joint angle trajectory $\tau$ as input, and predict a variable number of latent variables $\{z\}$, that correspond to the sequence of motor programs $\{M\}$ executed during trajectory $\tau$. We find the transformer architecture to be superior to LSTMs for processing long trajectories, due to its capacity to attend to parts of the trajectory as required.

Our abstraction network $\mathcal{A}$ predicts a varying number of primitives by additionally predicting a 'continuation probability' $p_k$ after each motor program variable $z_k$. We then predict an additional primitive only if the sampled discrete variable from $p_k$ is 1, and therefore also need to learn the prediction of these probabilities. While the loss function above can directly yield gradients to the predicted motor program encoding $z_k$ via $\mathcal{M}$, we use gradients using REINFORCE Williams (1992) (with $\Delta(\tau_{obs}, \tau_{rec}) + L_{sm}(\tau_{rec})$ as negative reward) to learn prediction of $p_k$.

## 3.2 ENFORCING SIMPLICITY AND PARSIMONY

While the objective described so far can in principle allow us to jointly learn the space of motor programs and understand the demonstration trajectories as a composition of these, there are additional properties we would wish to enforce to the bias the learning towards more desirable solutions. As an example, our framework presented so far can allow a solution where each demonstration is a motor program by itself *i.e.* the abstraction network can learn to map each demonstration to a unique $z$, and the primitive decoder can then decode this back. However, this is not a suitable solution as the learned motor programs are not 'simple'. On the other extreme, a very simple notion of a motor program is one that models each transition independently. However, this is again undesirable as this does not 'abstract' away the details of control or represent the demonstration as a smaller number of motor programs. Therefore, in addition to enforcing that the learned motor programs recompose the demonstrations, we also need to enforce simplicity and parsimony of these motor programs.

We incorporate these additional biases by adding priors in the objective or model space. To encourage the abstraction model to learn parsimonious abstractions of the input demonstrations, we penalize the number of motor primitives used to recompose the trajectory, by adding a small constant to the negative reward used to train the continuation probability $p_k$ if the corresponding sample yielded an additional primitive. To enforce simplicity of motor primitives, we observe that the trajectories yielded by a classical planner (in our case, RRT-Connect) *e.g.* to go from an initial to final state are 'simple' and we therefore initialize the motor primitive network using the decoder of a pretrained autoencoder on random planner trajectories for (start, goal) state pairs. Note that this notion of 'plannability' as 'simplicity' is merely one plausible alternative, and alternate ones can be explored *e.g.* 'linearity' (Kroemer et al., 2015) or 'predictability of motion' (Wolpert & Kawato, 1998).

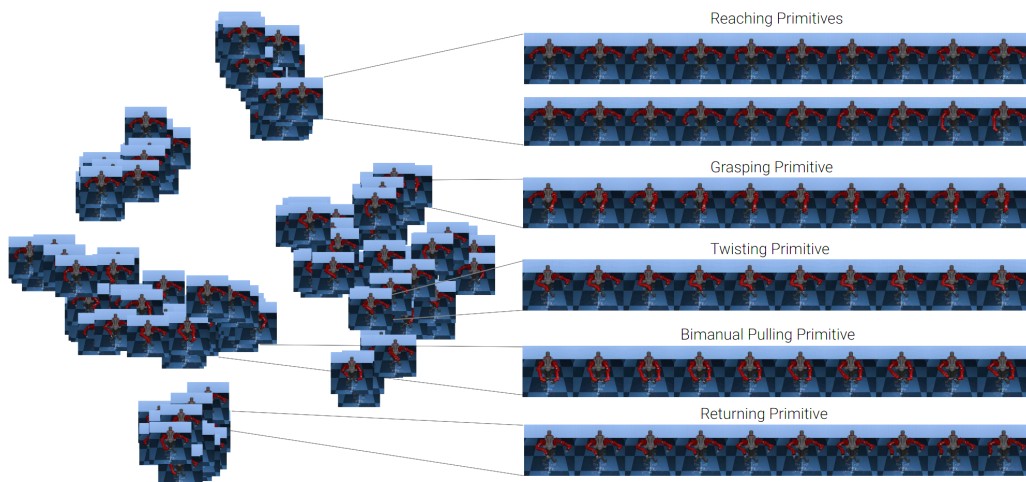

**Figure 3:** Visualization of the embedding of the latent representation of motor programs learned by our model (depicted on the left are the start configurations of each motor program, displayed at the corresponding position in the embedded space), and a set of sample primitives unrolled in time (depicted on the right). Each row corresponds to one primitive, annotated with semantic labels of what this motor primitive resembles.

## 4 EXPERIMENTS

We would like to ascertain how well our approach is capable of achieving our objective of successfully discovering and learning a good representation of the space of motor programs. Further, we seek to verify whether despite being learned in an unsupervised manner without semantic grounding, the learned primitive space is semantically meaningful. We would also like to evaluate how well they can be used to solve downstream RL tasks. We first provide a description of the data we wish to learn primitives from, followed by describing our quantitative and qualitative experiments towards verifying these three axes.

**Dataset:** We use the MIME dataset (Sharma et al., 2018) to train and evaluate our model. The dataset consists of over 8000 kinesthetic demonstrations of 20 tasks (such as pouring, pushing, bottle opening, stacking objects, etc.) collected on a real-world Baxter Robot. While the dataset has head and hand-mounted RGBD data, we use the Baxter joint angle trajectories to train our model. We consider a 16 dimensional space as our input and prediction space, consisting of 7 joints for each of the 2 arms, along with a scalar value for each gripper (we ignore torso and head joints). We emphasize this is a higher dimensional domain than most other related works consider. The gripper values are re-scaled to a $0-1$ range, while the joint angles are unnormalized. We temporally down-sample all joint angle data by a constant factor of 20 from the original data frequency of 100 Hz.

We randomly sample a train set of 5900 demonstrations from all 20 tasks, with a validation set of 1600 trajectories, and a held-out test set of 850 trajectories. To help evaluate the learned motor programs, we manually annotate a set of 60 test trajectories (3 trajectories from each task) with temporal segmentation annotations, as well as semantic labels of 10 primitives (such as reaching, twisting, pushing, etc.) that occur in these 60 trajectories. Note that these labels are purely for evaluation, our model does not have access to these annotations during training.

### 4.1 VISUALIZING THE SPACE OF PRIMITIVES

We would first like to evaluate the quality of the learned abstractions in and of themselves, i.e. *Is our approach able to discover the space of motor programs, and learn a good representation of this space?* We qualitatively answer this question by visualizing the latent representation space of motor primitives learned by our model. We first randomly sample a set of 500 trajectories unseen during training, then pass these trajectories through our model, and retrieve the predicted latent variables $\{z\}$ for each of these trajectories and their corresponding movement sequences $\{\bar{\tau}\}$. We then embed the latent variables in a 2-dimensional space using T-SNE (van der Maaten & Hinton, 2008), and visualize the corresponding movement sequences at their corresponding position in this 2-D embedded space, as in Fig. 3. We provide a GIF version of Fig. 3 (and other visualizations) at https://sites.google.com/view/discovering-motor-programs/home.

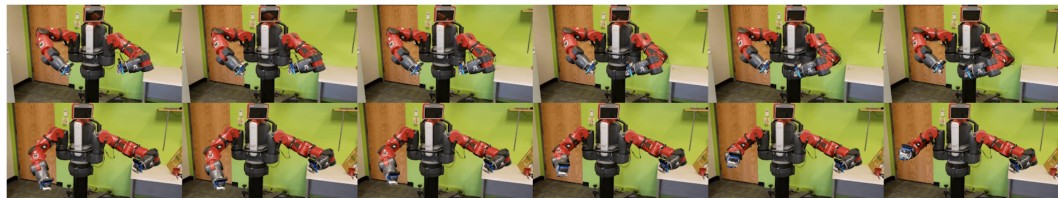

**Figure 4:** Depiction of execution of the learned primitives on real world Baxter robot. Each row is a single primitive, while columns show progress of the primitive over time. Row 1 depicts a left handed reaching primitive, while row 2 shows a right handed returning primitive. More visualizations provided in supplementary material, and videos are provided in the webpage.

We observe that clusters of movement sequences emerge in this embedded space based on the relative motions executed during the course of these trajectory segments (similar latent variables correspond to similar movement sequences and vice versa). While these clusters are not explicitly semantically labelled by our approach, the motions in these clusters correlate highly with traditional notions of skills in robot manipulation, i.e. reaching motions (top and top-left clusters), returning motions (bottom cluster), bi-manual motions (visible to the bottom right of the left most cluster and the bottom of the right most cluster), etc. We visualize a few such primitives (reaching, twisting, grasping, bi-manual pulling, etc.) among these to the right of Fig. 3. This shows our model learns smooth mappings $\mathcal{M}$ and $\mathcal{A}$, and is capable of *discovering* such primitives in an unsupervised manner without explicit temporal segmentation or semantic labels, which we believe is an encouraging result.

Interestingly, the model learns abstractions that pick up on the *trend* of the motion, rather than distinguishing between whether the left or right hand is used for the motion. This is particularly notable in the case of reaching and returning motions, where both left and right-handed reaching and returning motions appear alongside each other in their respective clusters in the embedded space.

### 4.1.1 Executing Primitives on a real Robot

We would ideally like the learned primitives from our model to be useful on a real Baxter robot platform, and be suitably smooth, feasible, and correspond to the motions executed in simulation (i.e. be largely unaffected by the noise of execution on a real robot). To verify whether our model is indeed able to learn such primitives, we execute a small set of learned primitives on a real Baxter robot, by feeding in the trajectory predicted by the model into a simple position controller. We visualize the results of this execution in Fig. 4, (see project webpage for videos). Despite not explicitly optimizing for feasibility or transfer to a real robot, the use of real-world Baxter data to train our model heavily biases the model towards primitives that are inherently feasible, relatively smooth and can be executed on a real robot without any subsequent modifications.

### 4.2 Semantic Segmentation Transfer using Learned Abstractions

As the 'recomposed' trajectory can be aligned to the original demonstration via sequence alignment, our predicted abstractions induce a partitioning of the demonstrated trajectory (corresponding to the aligned boundaries of the predicted primitives). We test whether the predicted abstraction and induced partitions are consistent across different demonstrations from the same task. To this end, we select 3 instances of the "Drop Object" task (Sharma et al., 2018) from the annotated test set, and retrieve the induced segmentations of the demonstration. We then visualize these segmentations and motor programs predicted for each of these 3 demonstrations as depicted in Fig. 5, along with the ground truth semantic labels of primitives for each of the demonstrations.

The alignment between a recomposed trajectory and the original demonstration also allows us to transfer semantic annotations from a demonstration to the predicted primitives, by simply copying labels from the demonstration to their aligned timepoints in the primitives. Therefore using our small *set* of annotated demonstrations, we can construct a small library of semantically annotated primitives. Given a novel, unseen demonstration, we can compute its predicted primitives and assign each a semantic label by copying the label of the nearest primitive from the library. This allows us to transfer semantic segmentations from our small annotated test set to unseen demonstrations. Our model's transfer of semantic segmentation achieves label accuracy on the set of 30 held out

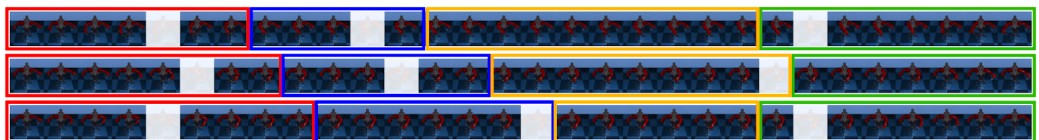

**Figure 5:** Visualization of consistent segmentations. Each row represents a different instance of a "Drop Objects" task from the MIME Dataset, while each column represents a time-step in the demonstration. White frames represent predicted segmentation points, while colored boxes represent ground truth semantic annotations. Red boxes are reaching primitives, blue boxes are grasping, orange boxes are placing, and green boxes are returning primitives. We see our model predicts 4 motor programs - reaching, grasping, placing the object a small distance away, and returning. This is consistent with the true semantic annotations of these demonstrations, and the overall sequence of primitives expected of the "Drop Box" task.

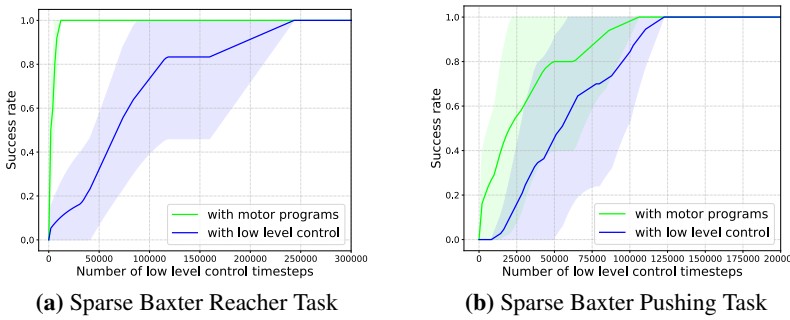

**(a)** Sparse Baxter Reacher Task          **(b)** Sparse Baxter Pushing Task

**Figure 6:** RL training curves with and without motor programs. Solid lines denote mean success rate, while shaded region denotes $\pm 1$ standard deviation across 10 random seeds.

trajectories (across all 20 tasks) of 58%, while a *supervised* LSTM baseline that predicts semantic labels from trajectories (trained on 30 annotated test trajectories) achieves 54% accuracy.

The consistency of our abstractions coupled with the ability to transfer semantic segmentations across demonstrations shows our model is capable of understanding commonalities across demonstrations, and is able to reason about various demonstrations in terms of motor programs shared across them, despite being trained in an unsupervised manner.

### 4.3 COMPOSING PRIMITIVES FOR HIERARCHICAL RL

One of our primary motivations for learning motor programs is that they can be composed together to solve downstream robotic tasks. To evaluate whether the motor programs learned by our model are indeed useful for such downstream tasks, we adopt a hierarchical reinforcement learning setup (as described in detail in the supplementary). For a given task, we train a policy to predict the sequence of motor programs to execute. Given the predicted latent representations, each motor program is decoded into its corresponding motion sequence using the previously learned motor program network. We retrieve a sequence of desired joint velocities from this motion sequence and use a joint velocity controller to execute these "low-level" actions on the robot. The motor program network and the joint velocity controller together serve as an "abstraction" of the low-level control that is executed on the robot. Hence, the policy must learn to predict motor programs that correspond to motion sequences useful for solving the task at hand.

As demonstrated in Fig. 6, training a policy using motor programs is several orders of magnitude more efficient than training with direct low-level actions. For the sparse reaching task, the motor program policy learns within 50 motor program queries, 2 orders of magnitude speedup in low-level control time-steps. For the sparse pushing task, the motor program policy learns within 1000 motor program queries, or a 2X speedup with respect to low-level control time-steps. We note that executing a motor program corresponds to 50 low level control steps (see appendix for details). However, as these motor programs are executed without environment feedback, the improvement in efficiency in terms of environment interactions is all the more significant.

## 5  CONCLUSION

We have presented an unsupervised approach to discover motor programs from a set of *unstructured* robot demonstrations. Through the insight that learned motor programs should recompose into the original demonstration while being simplistic, we discover a coherent and diverse latent space of primitives on the MIME (Sharma et al., 2018) dataset. We also observed that the learned primitives were semantically meaningful, and useful for efficiently learning downstream tasks in simulation. We hope that the contributions from our work enable learning and executing primitives in a plethora of real-world robotic tasks. It would also be interesting to leverage the learned motor programs in context of continual learning, to investigate how the discovered space can be adapted and expanded in context of novel robotic tasks.

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

## A APPENDIX

### A.1 EXECUTING PRIMITIVES ON A REAL ROBOT

We provide additional visualizations of primitives being executed on the real robot below. As mentioned in the main paper, dynamic GIFs of these visualizations may be found at https://sites.google.com/view/discovering-motor-programs/home.

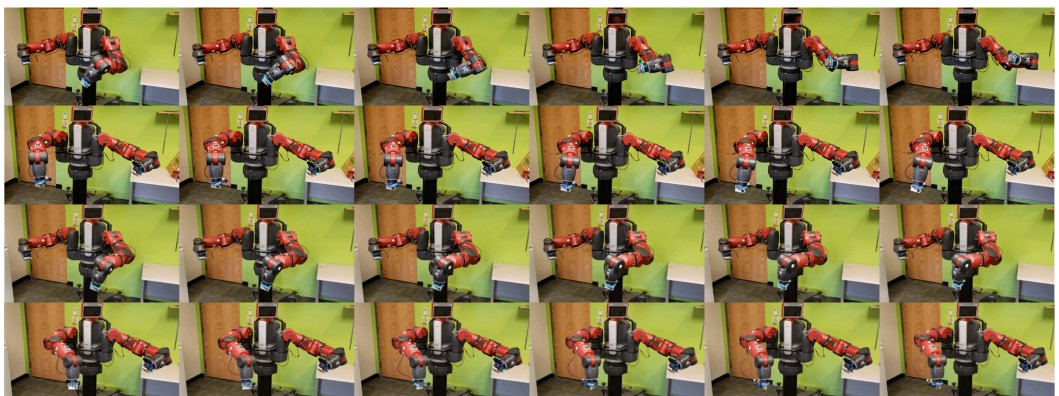

**Figure 7:** Depiction of execution of additional learned primitives on real world Baxter robot. As in Fig. 4, each row is a single primitive, while columns show progress of the primitive over time. Row 1 depicts a left handed returning primitive, row 2 depicts a right handed pushing primitive, row 3 depicts a left handed pushing primitive (in a different configuration to the left handed one), and finally row 4 depicts a right handed twisting primitive.

Our results webpage https://sites.google.com/view/discovering-motor-programs/home also contains visualizations of predicted *combinations* of primitives being executed on the real robot (as well as in simulation), along with corresponding demonstrations for comparison. We note that both the individual primitives, as well as the combinations of primitives are executed quite smoothly and naturally, and lead to motions that correspond highly to the original demonstrations. This indicates such combinations of primitives can indeed be used towards downstream tasks on the real robot.

### A.2 DETAILS OF HIERARCHICAL REINFORCEMENT LEARNING SETUP

For our hierarchical reinforcement learning experiments, we perform policy learning on two sparse reward (Andrychowicz et al., 2017) tasks on a simulated Baxter Robot: (a) Reaching, and (b) Pushing. For the reaching task, the robot's end-effector needs to reach a specific location in space, while for the pushing task, the robot needs to push a block on the table to a specific location.

For Baxter-Reaching task, the goal is to get the right-hand's end-effector to a pre-defined goal (x,y,z) state. The reward is a sparse reward with epsilon=0.05m; So if the end-effector reaches within 5 cm of the goal, it gets a reward of +1, otherwise it gets a reward of 0. For Baxter-Push, the goal is to get a block (cube) to the desired goal with epsilon=0.05m. To do this, the robot needs to hit/push the block to the goal. There is no other dense reward to encourage the robot to hit the block.

We train both our motor program policy and the baseline control policy using Proximal Policy Optimization (Schulman et al., 2017). Note that the motor program policy outputs the latent representation $z$. Each $z$ expands into a `10` length trajectory according the motor program network. To reach each of these trajectory states, a PD velocity controller is used for `5` time-steps. The baseline control policy directly outputs the velocity control action.

We note that our PPO baseline implements the same action for 10 timesteps, in a manner similar to frame-skipping, as is common in RL. We found that varying the number of timesteps frame-skipping was applied for did not have a significant effect on the performance of PPO.

