# OpenReview forum: "Discovering Motor Programs by Recomposing Demonstrations"
_ICLR.cc/2020/Conference — Accept (Poster)_

### Official Review · AnonReviewer1 · 2019-10-22
**Official Blind Review #1**

**Rating:** 8

**Review:**

This work presents a novel approach to extracting reusable motor primitives from task demonstrations.  The approach taken in this work involves learning a deep encoder network which translates an arbitrary length trajectory in a robot's configuration space (is this right?) into a sequence of vectors describing different motor primitives.  A second decoder network translates these vectors into a sequence of trajectory segments.  These networks are trained to minimize the distance between the original trajectory, and the trajectory generated by encoding and reconstructing the original as a sequence of primitives and reconstructing.  An additional regularization term discourages the network from learning trivial, one step primitives.  The decoder network is also initialized by training on a set of simple trajectories generated by a robotic planning algorithm.

Experiments involved extracting motor programs from the MIME data set consisting of demonstrated trajectories for the Baxter robot.  In addition to qualitative visualizations of the learned primitives, quantitative results using a limited set of human-segmented trajectories demonstrate that the learned primitives roughly correspond to the segmentations that humans identify.  Further experiments show that reinforcement learning in the space if learned primitives is more sample efficient than RL in the low-level control space.

The work presents a novel and effective solution to the difficult task of learning reusable motor skills.  While the work focuses on robotic control, it is likely that similar approaches could be developed for more general reinforcement learning problems.  There is room for improvement.  In particular, ablations on the regularization and initialization mechanisms could help us better understand the importance of these elements in learning useful motor programs, and illustrate the robustness of this method to different methods of initialization.

**Experience Assessment:**

I have read many papers in this area.

**Review Assessment: Checking Correctness Of Derivations And Theory:**

I assessed the sensibility of the derivations and theory.

**Review Assessment: Checking Correctness Of Experiments:**

I assessed the sensibility of the experiments.

**Review Assessment: Thoroughness In Paper Reading:**

I read the paper at least twice and used my best judgement in assessing the paper.

---

> ### Author Response · Authors · 2019-11-13
> **Response to Reviewer #1**
>
> We thank the reviewer for their appreciation of the novelty and effectiveness of our approach, and noting our method is generally more applicable beyond the robotics domain.
>
> Regarding the initialization of our networks -
> We observed the initialization of the skill network by imitating planner trajectories to be necessary for our approach to function. Without a meaningful initialization like this (or alternatively, something like autoencoding trajectory segments from the demonstration), the learning problem at hand is very challenging, as the decoding of latent variables into trajectory segments would change over time.
>
> Regarding the regularizations of simplicity, parsimony, and smoothness losses -
> We note that the use of smoothness loss is not critical to learning. Our model does learn a reasonable latent space without the smoothness loss, however the  selected primitives often result into large, discontinuous state changes between boundaries of primitives.
> The use of simplicity (by means of the initialization) was critical to learning, as otherwise the network simply learnt to encode the entire trajectory in one latent variable. Conversely, without the parsimony regularization and the continuation-probability bias (as noted in section 3.2), the network learnt to predict primitives of one timestep each.
> In summary, without the use of these regularization methods and initializations, the overall learning collapses to degenerate solutions.
>
> Regarding Configuration Space -
> Yes, we demonstrate our approach in the robot configuration space, but note that it may also be readily applied to the end effector space (although a suitable change of the distance function to incorporate orientations may be required).

---

> > ### Comment · AnonReviewer1 · 2019-11-14
> > **Experimental results**
> >
> > Thank you for taking the time to address these comments.
> >
> > I am still a little unclear on how the baseline (low-level control) learner used in section 4.3 is defined.  As I understand it, what is being learned in this case is just a flat control policy mapping from robot configurations to joint torques, which as reviewer 4 pointed out, may define a different, and more challenging RL problem.
> >
> > It could be argued that reducing the complexity of the RL problem is the point of learning primitives in the first place, but more details about how this comparison is being conducted (e.g., what RL algorithm is being used, what is the reward signal) would be imensely helpful in understanding the significance of these results.
> >
> > It would also seem that a natural alternative basline would be training the full encoder, decoder network end-to-end without any pretraining of the decoder.  This would clearly demonstrate that you are actually transferring previously learned skills in a way that leads to improved learning performance on the target tasks.

---

> > > ### Author Response · Authors · 2019-11-15
> > > **Response to Reviewer #1's comment**
> > >
> > > We hope the following clarifications help in understanding the significance of our RL results.
> > >
> > > In the case of the low-level control baseline policy, we train a Multi-layer perceptron policy to predict joint velocities, given the robot configuration as input. As is done in other deep RL algorithms, this low-level control baseline executes the same action for 10 timesteps, in a manner similar to frame-skipping.
> > >
> > > In contrast, our RL approach trains a policy to predict a sequence of 5 latent z vectors, that each expand into a 10 length trajectory according the motor program network. To reach each of these trajectory states, a simple Proportional controller is used for 5 time-steps.
> > > We train both our motor program policy and the baseline control policy using Proximal Policy Optimization [1].
> > >
> > > As mentioned in our response to Reviewer #4 above, the details of the reward signal used are as follows. For Baxter-Reaching task, the goal is to get the right-hand’s end-effector to a pre-defined goal (x,y,z) state. The reward is a sparse reward with epsilon=0.05m; So if the end-effector reaches within 5 cm of the goal, it gets a reward of +1, otherwise it gets a reward of 0. For Baxter-Push, the goal is to get a block (cube) to the desired goal with epsilon=0.05m. To do this, the robot needs to hit/push the block to the goal, and gets a reward of +1 upon reaching this goal, and 0 otherwise. There is no other dense reward to encourage the robot to hit the block.
> > >
> > > Regarding the baseline suggested by the reviewer:
> > > We train an encoder decoder style policy in a setting identical to our approach, except without pretraining the decoder (i.e. the abstraction network). We try two variants:
> > >
> > > 1) Keep Decoder Random and Fixed: As expected this does not train at all. This highlights that a  random decoder, unlike our pre-trained one, does not give a meaningful skill space.
> > >
> > > 2) Train Decoder along with Encoder: Unfortunately, learning to predict z's and decoding into trajectories jointly is hierarchical policy learning in the RL setting, and this is really hard in the sparse reward case. Consequently, we found that even this baseline fails to train. We also note that even if this succeeds, it implies having a different ’skill space’ per downstream task, which is undesirable.
> > >
> > > [1]  J. Schulman, F. Wolski, P. Dhariwal, A. Radford, and O. Klimov. Proximal policy optimization algorithms. https://arxiv.org/abs/1707.06347

---

> > > > ### Comment · AnonReviewer1 · 2019-11-15
> > > > **Response**
> > > >
> > > > Thank you for providing these details, I believe this addresses my concerns.  Please make sure that they are included in the paper, or in the supplementary material.

---

### Official Review · AnonReviewer3 · 2019-10-26
**Official Blind Review #1246**

**Rating:** 6

**Review:**

The paper aims to learn middle-level motor task primitives from unlabeled actions. The main insight is that the decomposition of motor tasks can be learned using a set of LSTMs with a loss function that minimizes the differences between the original task and the recomposed task. They evaluate their approach on MIME dataset that includes 20 different tasks.

+ The idea of recomposition based loss function seems very useful for learning from unlabeled data.
+ The evaluation results seem to be strong. It outperforms a supervised LSTM baseline by 4 percentage points.

- The related work is somewhat narrowly focused on the controlled program. It will be nice if the authors can describe whether such ideas have explored in other domains before.

- It is not clear to me how much the accuracy gain in the latent representation transfer to the accuracy of the actual recomposed task. The authors presented no quantitative results to show how much the 4pp gain improves the accuracy of new tasks.



**Experience Assessment:**

I do not know much about this area.

**Review Assessment: Checking Correctness Of Derivations And Theory:**

I did not assess the derivations or theory.

**Review Assessment: Checking Correctness Of Experiments:**

I assessed the sensibility of the experiments.

**Review Assessment: Thoroughness In Paper Reading:**

I made a quick assessment of this paper.

---

> ### Author Response · Authors · 2019-11-13
> **Response to Reviewer #3**
>
> We thank the reviewer for their appreciation of our paper, and the use of recomposition based loss functions in unsupervised learning.
>
> Regarding similar ideas in other domains -
> There are indeed parallels between our approach and some generative modeling works in spatial domains (images/3D shapes) that similarly compose some notion of ‘primitives’ to explain the input. For example, [1] addresses image generation by recurrently “drawing strokes” for handwritten digits. [2] addresses the inference of a sequence of latent variables, in the context of both image and 3D scene generation. Like [1], it uses “strokes” to compose hand-written characters. In both [1] and [2], the “strokes” used may be interpreted as primitives, or simply as 2D trajectories. [3] bears similar high level ideas of learning abstractions via assembly. In particular, they seek to learn shape abstractions by assembling predefined volumetric primitives to coarsely reconstruct 3D shapes. We will include a brief summary in related work.
>
> [1] DRAW: https://arxiv.org/pdf/1502.04623.pdf
> [2] Attend Infer Repeat: https://arxiv.org/pdf/1603.08575.pdf
> [3] Learning Shape Abstractions by Assembling Volumetric Primitives: https://arxiv.org/pdf/1612.00404.pdf
>
> Regarding accuracy gain transfer to recomposition task -
> We note that the accuracy at semantic transfer, downstream RL, and recomposition actually evaluates three critical but complementary aspects respectively: a) that the primitives are semantically meaningful units, b) the primitives can be combined for efficiently learning downstream tasks, and c) we can capture the space of motions faithfully.

---

### Official Review · AnonReviewer4 · 2019-11-05
**Official Blind Review #4**

**Rating:** 3

**Review:**

Paper Summary:
The paper proposes a method for learning a set of primitives for robotic movements from a dataset of demonstrations, showing a diverse set of tasks, in an unsupervised fashion. The central underlying idea is that robotic tasks can be solved by combining fundamental building blocks, the so-called "motor programs", in the right way. The described algorithm takes a demonstration and uses a transformer network to embed the trajectory into a sequence of latent variables. Then each individual latent is transformed to a 10 step trajectory for the joint space of the robot via an LSTM network. Finally the individual trajectories are concatenated and the reconstruction is compared to the original demonstration through dynamic time warping. In this structure the latent variables represent a query to a specific learned primitive, which can be accessed using the LSTM.
In the experimental section, the paper gives mostly qualitative insight into the learned representation. The paper visualizes different movements and their corresponding latent variables projected onto two dimensions. Further, it is shown that the segmentation of demonstrations roughly compares to how a human expert would manually segment the given tasks. Finally, the paper shows that a hierarchical RL algorithm trained in the learned latent space outperforms one which works directly in the low-level control space of the robot in the sense that it learns to solve the given task much faster.

Evaluation:
The problem of discovering primitives is approached by the authors in a novel and interesting way, however in my opinion the paper should be rejected because:
    (a) the experimental section is not convincing enough to support the claim that the method captures the shared motions across different skills. Especially, the paper misses to adequately show how these motions can be recombined and used to solve robotic tasks.
    (b) the paper is imprecise and missing important details in both the description of the method and the experimental verification
    (c) the paper misses important related work, which tackles the same problem.


The two main claims in the paper are:
1. The presented method learns a latent space which represents common shared motions among diverse tasks encountered in robotics
2. Robotic tasks can be solved by recombining primitives from the aformentioned space

Although it is impossible to verify the first claim based on the paper alone, the provided webpage, which shows an animated version of Figure 3 nicely visualizes the learned latent space and shows that the representation is somewhat smooth with similar movements clustered together.

a1) Figure 5 is supposed to show that the method manages to segment given demonstrations in a meaningful way, but even after zooming into the pdf, it is impossible to see what is actually going on. A visualization in a video would be preferable.

a2) 4.1.1 and Figure 4 show that individual primitives can be executed on a real robot, however, the paper fails to show the execution of a combination of primitives.

a3) Given that the method seems to loose the connection of movements to time it would be interesting to see whether a combination actually results in smooth, natural movement of the robot.

a4) The main quantitative assessment of the usefulness of the learned motor program network is given by the RL experiments in section 4.3. However, the baseline method seems to output one single velocity control action per evaluation of the policy (?), whereas the presented method essentially outputs an action sequence of 50 actions per policy evaluation. State-of-the-art methods commonly use frame-skipping and repeat the same action for multiple timesteps, because it makes the resulting optimization problem easier and speeds up the learning. See for example (Mnih 2013) (Mnih 2015) (Lillycrap 2015) (Hafner 2018). It would be interesting to compare against a baseline which also incorporates some form of frame-skipping and validate the speedup is not simply due to chunking of action sequences.

a5) Finally, the method outperforms the plain PPO baseline when it comes to speeding up the learning process, but the solutions found do not look like natural robot movements. You can clearly spot different primitives and transitions between individual segments look unnatural and jerky. How does the baseline solution compare in this regard?

a6) Given that the presented movements look unnatural and the fact that the paper only shows the execution of individual primitives in the rest of the paper, I simply cannot support the second claim.


Details I am missing from the paper:
b1) How is the "continuation probability" computed with the transformer network?
b2) Are the biases in section 3.2 necessary to make the method work at all?
b3) In section 4.2, how does the sequence alignment work?
b4) In section 4.2, what is the training set for the labelling task?
b5) What are the exact task parameters given to the policy in the RL task?
b6) What do you mean with, "these motor programs are executed without environment feedback"? Does the policy determine the complete sequence of programs in one step?


c1) Finally, to my knowledge the problem of learning meaningful primitives and showing that they can be combined in a different way to solve novel, unseen tasks has already been investigated in (Lioutikov 2017). Although this paper approaches the problem very differently and the dimensionality of the primitives is lower, I still consider this paper very much related to what is presented by the authors. I would suggest adding it to the related work. The paper shows that previous methods for discovering primitives from a set of different tasks exist.


In my opinion the paper in its current is probably not yet ready for publication. However, I strongly encourage the authors to address the above mentioned problems.

References:
Mnih, Volodymyr, et al. "Playing atari with deep reinforcement learning." arXiv preprint arXiv:1312.5602 (2013).
Mnih, Volodymyr, et al. "Human-level control through deep reinforcement learning." Nature 518.7540 (2015): 529.
Lillicrap, Timothy P., et al. "Continuous control with deep reinforcement learning." arXiv preprint arXiv:1509.02971 (2015).
Hafner, Danijar, et al. "Learning latent dynamics for planning from pixels." arXiv preprint arXiv:1811.04551 (2018).
Lioutikov, Rudolf, et al. "Learning movement primitive libraries through probabilistic segmentation." The International Journal of Robotics Research 36.8 (2017): 879-894.

**Experience Assessment:**

I have read many papers in this area.

**Review Assessment: Checking Correctness Of Derivations And Theory:**

I assessed the sensibility of the derivations and theory.

**Review Assessment: Checking Correctness Of Experiments:**

I carefully checked the experiments.

**Review Assessment: Thoroughness In Paper Reading:**

I read the paper at least twice and used my best judgement in assessing the paper.

---

> ### Author Response · Authors · 2019-11-13
> **Response to Reviewer #4 (Part 1)**
>
> Please note: We split our response across two comments due to character limit.
>
> We thank the reviewer for the detailed review, and address the concerns raised in the following manner:
> Regarding showing the recombination of the learned primitives, we have added additional results of the combinations of primitives to produce entire trajectories, executed both in simulation as well as on the real robot. We hope this strongly supports the second claim and highlights that our primitives can indeed be combined to solve robotic tasks.
> Regarding the missing details pointed out, we provide these details below, and shall add them into our paper.
> We note how our approach is related to the mentioned paper [Lioutikov et. al., 2017], and shall add this to our body of related work.
>
> Regarding (A) comments:
> A2) We have updated our results webpage (https://sites.google.com/view/discovering-motor-programs/home#h.p_7TW5Dc4GshZ6) with visualizations of predicted combinations of primitives being executed on the real robot (as well as in simulation), along with corresponding demonstrations for comparison.
> We note that both the individual primitives, as well as the combinations of primitives are executed quite smoothly and naturally, and lead to motions that correspond highly to the original demonstrations. We hope this allays concerns the reviewer had regarding combinations of primitives being lacking, and convinces the reviewer that such combinations of primitives can indeed be used towards downstream tasks on the real robot.
>
> A3) The reviewer raises the interesting point of capturing time in the primitives. While we believe this is necessary to address general dynamic tasks, the tasks that appear in our demonstrations (and hence the primitives we learn) are not dynamic tasks but quasi-static. Since our primitive representation is learnt jointly over all joints, synchronizing various joint motions is not an issue. We hence believe it is sufficient to have a primitive representation that captures the shape of motions executed.
>
> Further, as our original visualizations of primitives on the real robot showed, the learnt primitives are fairly smooth motions in and of themselves even with a simple P-controller being used. This results from being trained on inherently smooth demonstration data.
>
> The updated visualizations of combinations of primitives in our webpage further show that the combinations of primitives can also be executed smoothly, without jumps or discontinuities between primitives, due to the smoothness loss used. The combinations of primitives visualized appear to reflect their corresponding demonstrations well, and result in natural appearing trajectories that could be executed to achieve respective tasks.
>
> A4) We note that our PPO baseline also implements the same action for 10 timesteps, in a manner similar to frame-skipping mentioned by the reviewer. We found that varying the number of timesteps frame-skipping was applied for did not have a significant effect on the performance of PPO.
>
> A5) We did observe that the baseline solution of PPO while being inefficient does produce smoother trajectories, since it locally selects small actions from consecutive states. In contrast, the jerkiness observed in our approach is due to discontinuities between end and start of consecutive primitives, and can be mitigated for our approach by simply incorporating a smoothness loss during RL training (as was used in Sec 3.2 to train our abstraction network), although we did not include this term in RL training for simplicity.
>
> A6) We hope that the additional visualizations mentioned above (i.e. the combinations of primitives on the real robot and their natural appearance) convinces the reviewer of our claims, and helps make obvious the significant potential to solve tasks by composing primitives produced by our approach.

---

> > ### Author Response · Authors · 2019-11-13
> > **Response to Reviewer #4 (Part 2)**
> >
> > Regarding missing Details requested ([B] Comments):
> > B1) The continuation probability are predicted by use of a softmax layer over two classes (corresponding to continuing the current primitive, or starting a new primitive). This is simply predicting a Bernoulli distribution at every timestep.
> >
> > B2) We reiterate our comment in reply to Reviewer #1 below regarding the necessity of regularizations:
> > The use of smoothness loss is not critical to learning. Our model does learn a reasonable latent space without the smoothness loss, however the  selected primitives often result into large, discontinuous state changes between boundaries of primitives. The use of simplicity (by means of the initialization) was critical to learning, as otherwise the network simply learnt to encode the entire trajectory in one latent variable. Conversely, without the parsimony regularization and the continuation-probability bias (as noted in section 3.2), the network learnt to predict primitives of one timestep each. In summary, without the use of these regularization methods and initializations, the overall learning collapses to degenerate solutions.
> >
> > B3) The sequence alignment mentioned in section 4.2 is the same sequence alignment alluded to in section 3.1 (in particular, Equation 3).
> > It works by using Dynamic Time Warping to find a set of valid indices (subject to constraints mentioned in section 3.1) that map from one sequence to another and vice versa, that incurs minimum cost as given by equation 3. The alignment is given by this set of indices.
> > The segmentation transfer mentioned in section 4.2 simply selects labels from an annotated trajectory, and transfers them to target trajectory via this set of indices.
> >
> > B4) As described in the beginning of section 4, we manually annotate 60 trajectories across the 20 tasks in the dataset. 30 randomly selected trajectories out of these 60 annotated trajectories are used as the training set, while the remaining 30 serve as the test set.
> >
> > B5) For Baxter-Reaching task, the goal is to get the right-hand’s end-effector to a pre-defined goal (x,y,z) state. The reward is a sparse reward with epsilon=0.05m; So if the end-effector reaches within 5 cm of the goal, it gets a reward of +1, otherwise it gets a reward of 0. For Baxter-Push, the goal is to get a block (cube) to the desired goal with epsilon=0.05m. To do this, the robot needs to hit/push the block to the goal. There is no other dense reward to encourage the robot to hit the block. For both the environments the input to the policy is the robot configuration (joint angles), while actions are specified as joint velocities of the robot. For executing primitives, a Proportional controller is used for primitive trajectory tracking.
> >
> > B6) The motor programs are executed in an open-loop, without perceptual feedback affecting the trajectory once it is selected. The policy uses perceptual feedback at each of it’s 5 steps, to select a motor program to execute.
> >
> >
> > Regarding Related Work ([C] comments):
> > C1) We thank the reviewer for pointing out the work of Lioutikov et. al. [1]. We agree that it is, at a high level, also addressing learning a primitive representation alongside the segmentations of a demonstration, and shall accordingly note this in our body of related work.
> > We would, however, like to emphasize that our work emphasizes learning of such a representation across a large, diverse set of demonstrations, so as to retrieve a correspondingly diverse set of skills. In particular, the library that [1] provides is a finite set, which greatly restricts its applicability to the large scale set of skills we hope to retrieve. However, by maintaining a continuous parameterization of the space of skills, we are able to bypass the difficulty of a fixed size library.
> >
> > [1] Lioutikov, Rudolf, et al. "Learning movement primitive libraries through probabilistic segmentation." The International Journal of Robotics Research 36.8 (2017): 879-894.

---

> > > ### Comment · AnonReviewer4 · 2019-11-14
> > > **Response to Authors of 1246**
> > >
> > > Thank you for addressing the remarks and providing additional details as well as further visualizations on your results webpage.
> > >
> > > I have a quick question regarding the additional visualizations of the combination of primitives. I assume you take a demonstration from the dataset, use the transformer network to embed it into the latent space and then use the resulting latents to reconstruct the demonstration with the LSTM (Motor Network). Afterwards the 10-step programs are executed with a proportional controller on the joint positions. How do you handle the time in this case? (Is the proportional controller simply active for a given time, or until the target is reached, etc.)

---

> > > > ### Author Response · Authors · 2019-11-14
> > > > **Response to Reviewer #4 Comment**
> > > >
> > > > Yes, we decode latents retrieved from the transformer to showcase the combination of primitives. This is so we can compare the resultant trajectories with their corresponding demonstrations.
> > > >
> > > > Regarding the handling of time for execution of trajectories - The controller is indeed run for a fixed amount of time. This time of execution (set roughly to 1 second) was determined to be large enough that the controller can reach close to the target state (or there would be large, jerky motions when the target state is changed to a subsequent state), while still being small enough to ensure the arm executes a continuous motion.
> > > >
> > > > We hope this answers the reviewer's question.

---

### Decision · Program_Chairs · 2019-12-19

**Decision:**

Accept (Poster)

**Comment:**

The work presents a novel and effective solution to learning reusable motor skills.  The urgency of this problem and the considerable rebuttal of the authors merits publication of this paper, which is not perfect but needs community attention.